# The Role of Work Engagement in the Association between Psychological Capital and Safety Citizenship Behavior in Coal Miners: A Mediation Analysis

**DOI:** 10.3390/ijerph18179303

**Published:** 2021-09-03

**Authors:** Kuiyuan Qin, Zhaona Jia, Tianjiao Lu, Saifang Liu, Jijun Lan, Xuqun You, Yuan Li

**Affiliations:** 1School of Psychology, Shaanxi Normal University, Xi’an 710061, China; qinkuiyuan@163.com (K.Q.); jzn2019@snnu.edu.cn (Z.J.); liusaifang0618@163.com (S.L.); spchild@163.com (J.L.); 2Shaanxi Provincial Key Laboratory of Behavior and Cognitive Neuroscience, Xi’an 710061, China; 3Student Mental Health Education Center, Northwestern Polytechnical University, Xi’an 710072, China; ltj2003@nwpu.edu.cn

**Keywords:** psychological capital, work engagement, safety citizenship behavior, coal miners

## Abstract

With the development of science and technology and the increasing importance attached by to these domains by the state and government departments in recent years, China’s coal production and safety supervision level continue to increase. However, the prevalence of frequent coal mine safety accidents has not been effectively curbed. The main purpose of this study was to explore the mediating role of work engagement in the relationship between psychological capital and safety citizenship behavior among Chinese coal miners. Data for 317 coal miners were collected from five coal and energy enterprises. The Psychological Capital Questionnaire (PCQ), the Job Engagement Scale (JES), and the Safety Citizenship Behavior Scale (SCBS) were used to evaluate the coal miners’ psychological capital, work engagement, and safety citizenship behavior. The causal steps approach and bootstrap Method were used in this study to assess the proposed mediation models. A correlation analysis indicated that psychological capital, work engagement, and safety citizenship behavior were significantly correlated with each other. Furthermore, the mediation analysis showed that work engagement mediated the relationship between psychological capital and safety citizenship behavior. Psychological capital does not only have a direct impact on coal miners’ safety citizenship behavior, but it also has an indirect impact on coal miners’ safety citizenship behavior via work engagement. Therefore, effectively enhancing an individual’s psychological capital and work engagement may be a basic factor determining coal miners’ safety citizenship behavior, which further promotes safety production within the enterprise.

## 1. Introduction

In China, being a coal miner is listed by the state as one of the high-risk professions due to its harsh working environment, complex operation tasks, long operation time, intense work, repetitive tasks and high workload [1]. In recent years, with the development of science and technology and increasing concern from the state and government departments, China’s coal production and safety supervision levels have continued to increase; however, the prevalence of frequent coal mine safety accidents has not been effectively curbed, and establishing and maintaining conditions of safety during production in the coal industry remain critical. According to the data of China Coal Accidents and Expert Comments, among all the direct causes of major accidents in China’s coal mines, the proportion of human factors is 97.67%—including 45.89% deliberate violations, 44.89% management errors, 0.69% design defects, and 8.53% other unintentional behavior—while the proportion of human factors’ responsibility in dust explosion, blasting, transport and lifting electromechanical accidents is almost 100%. Therefore, human errors are an important factor that both threatens the safety of coal miners and affects the stable development of the whole coal industry and national economic construction.

In the study of human errors in other high-risk occupational groups such as chemical plant workers and high-altitude construction workers [2,3,4,5], safety citizenship behavior is an important factor that influences their errors and safety performance. In other words, individual safety citizenship behavior can effectively promote or improve the work safety level of high-risk occupational groups. However, we ask ourselves the following questions of: what factors affect an individual’s safety citizenship behavior; what their specific mechanisms of action are for individual safety citizenship behavior; how can an individual’s safety citizenship behavior be effectively enhanced; and what measures and opinions may be provided. This study will take coal miners as an example to research and discuss these issues.

The concept of safety citizenship behavior is derived from organizational citizenship behavior, and it is a special kind of organizational citizenship behavior. Organizational citizenship behavior refers to the employee’s volunteer individual behavior beyond an organization’s formal remuneration system [6]. Such behavior can improve the performance of the organization as a whole but does not receive any direct or clear return from the organization’s remuneration system [7]. Organizational citizenship behavior is characterized by three aspects: first, it is not behavior required by the mandatory requirements of a job role or job description, therefore, if the individual does not display organizational citizenship behavior, he will not be punished; second, only long-term and repeated organizational citizenship behavior made by the staff can help enhance the performance of the organization, whilst occasional organizational citizenship behavior rarely has a great impact; third, although the organizational citizenship behavior obtains no direct or clear return from the organization’s formal remuneration system, it is possible for individuals who display organizational citizenship behavior receive return in other ways, such as their own safety [8]. Additionally, safety citizenship behavior is proposed on this basis, in reference to an individual’s volunteer behavior to ensure the safety of other team members and the safety performance of the project’s organization. This behavior cannot be directly or accurately identified and rewarded by the traditional reward system, but it can effectively enhance the safety of the entire organization’s work [9,10].

Safety citizenship behavior is significantly different from the safety behavior of traditional management psychology which refers to a kind of passive obedience of employees, emphasizing the regulation and restriction of safety laws or regulations on individual behavior in order to protect individual safety. Safety citizenship behavior emphasizes a kind of spontaneous and proactive safety behavior of individuals. In other words, the emergence of citizenship behavior research also marks that more and more research is shifting from the previous focus on the staff’s passive safety to the present focus on their positive safety.

## 2. Literature Review

### 2.1. Psychological Capital and Safety Citizenship Behavior

Psychological capital is an important concept in positive psychology, which refers to a positive psychological state presented by the individual in the process of growth and development [11]. It includes the four following aspects: (1) self-efficacy—the individual is confident and capable of making efforts to succeed in the face of challengeable work; (2) optimism—the individual attributes positivity to present and future success; (3) hope—the individual perseveres towards their goal and is capable of changing their means of achieving it; and (4) resilience—the individual can persevere in the face of adversity and trouble and quickly adjust their own state to achieve a breakthrough and ultimately succeed [12,13]. Psychological capital is known as the core capital of the fourth largest enterprise after economic capital, social capital and human capital in the modern market economy. Although the concept of psychological capital has only recently appeared, it has gradually become a very important and popular field in positive organizational behavior research. Moreover, a large number of studies have revealed that psychological capital can have a great impact on the individual’s work attitude and performance, especially in terms of organizational citizenship behavior [14,15,16,17,18]. Therefore, in a sense, psychological capital represents an important resource and means for individuals and organizations to obtain a sustainable competitive advantage in their future development. However, it is worth noting that most of the aforementioned studies focused on the impact of individual psychological capital on organizational citizenship behavior, while only a few focused on the close and stable relationship between psychological capital and safety citizenship behavior. Therefore, for this reason and on the above theoretical basis, this study focused on the relationship between the psychological capital and safety citizenship behavior of coal miners and proposed the following hypothesis:

**Hypothesis** **1****(H1).**
*Coal miners’ psychological capital can significantly predict their safety citizenship behavior.*


### 2.2. Work Engagement and Safety Citizenship Behavior

Work engagement means that the members of the organization fully embrace and dedicate themselves to their job roles through self-control [19,20]. This relationship shows that when an individual has high work engagement, they will put more energy into the work role and strive to show their best, but when the individual has low work engagement, they will tend to diverge from the job role, show lower job performance, and may even have the intention to leave [21]. Work engagement consists of physiological engagement, cognitive engagement and emotional engagement. Physiological engagement means that the individual can maintain a high degree of physiological involvement in the implementation of the role task; cognitive engagement means that the individual can maintain a high degree of cognition through an active and attentive state, and can be clearly aware of their role and mission in a particular work situation; emotional engagement means that the individual can maintain a connection with other people (such as colleagues and superiors) and sensitivity to the emotions and feelings of others [22]. These three aspects are relatively independent as, for example, although the individual maintains a high degree of physiological engagement, they may be inactive in terms of their cognition and emotion. However, in general, the higher the work engagement of the individual in a certain dimension, the higher the overall work engagement is. A large number of studies have shown that work engagement can have an important impact on the individual’s work attitude and performance, especially organizational citizenship behavior [23,24,25,26,27,28,29]. However, similarly to the studies of the relationship between psychological capital and safety citizenship behavior, most focus on organizational citizenship behavior to study the impact of work engagement on organizational behavior. For example, Runhaar and colleagues explored the relationship between work engagement and organizational citizenship behavior in a teachers’ group [24]. Lyu and colleagues tested the relationship between the two factors in hotel staff [27]. However, studies of the relationship between work engagement and safety citizenship behavior are not adequate. After 2018, researchers in different fields have begun to pay attention to the role of safety citizenship behavior in safety production. To date, we found few studies exploring the impact of work engagement on safety citizenship behavior, especially in coal mine safety production. Therefore, for this reason and on the above theoretical basis, this study attempted to discuss the relationship between the coal miners’ work engagement and safety citizenship behavior and proposed the following hypothesis:

**Hypothesis** **2****(H2).**
*Coal miners’ work engagement can significantly predict their safety citizenship behavior.*


### 2.3. Psychological Capital and Work Engagement

Compared to studies of safety citizenship behavior, research into the relationship between psychological capital and work engagement has been relatively well developed. A large number of studies have shown that individuals with higher psychological capital have a better work engagement state within the organization or at work than those with lower psychological capital [30,31,32,33]. Moreover, psychological capital both plays a vital role in the individual’s work engagement and has an important impact on job burnout—which a large number of studies have also revealed—meaning that individuals with higher psychological capital often show a lower job burnout tendency [34,35,36,37,38]. Job burnout is the direct opposite of work engagement, as individuals with high work engagement are energetic, can effectively enter their working state, get along with others, and be fully competent in fulfilling various work requirements; on the contrary, individuals with high job burnout display a sense of inefficiency and exhaustion and operate in an alienated state from work and others. Therefore, studies of the relationship between psychological capital and job burnout indirectly prove that the psychological capital of staff plays a very important role in the individual’s work engagement.

In a word, considering the above theoretical basis, there is a close relationship among the individual’s psychological capital, work engagement and safety citizenship behavior. Therefore, this study of the safety citizenship behavior of coal miners, a high-risk occupational group, is expected to reveal the specific mechanism of action and the model of connection between psychological capital, work engagement and safety citizenship behavior. Therefore, this study proposed the last two hypotheses:

**Hypothesis** **3****(H3).**
*Coal miners’ psychological capital can significantly predict their work engagement.*


**Hypothesis** **4****(H4).**
*Coal miners’ work engagement mediates the relationship between psychological capital and safety citizenship behavior.*


By reviewing former research, there are limited studies on the relationship among psychological capital, work engagement and safety citizenship behavior in coal miners. Specifically, the mediating role of work engagement between psychological capital and safety citizenship behavior is rarely mentioned. Thus, this research focused on the human factors of coal miners, applying a structural equation model to analyze the relationship among psychological capital, work engagement and safety citizenship behavior, and further test the mediating role of work engagement.

## 3. Materials and Methods

### 3.1. Participants

This study was conducted in June 2019. A total of 317 coal miners (valid data samples) from 5 coal and energy enterprises in Shaanxi, Inner Mongolia, and Hebei in mainland China volunteered to participate in this study. Coal miners were made aware of the survey via advertisements on noticeboards. All participants were male, and their mean age was 31.14 years (SD = 4.46). Participation was anonymous, and they were told that they were free to withdraw from the study at any time. All participants completed the three measures in a quiet environment. All procedures were executed in compliance with the relevant laws and institutional guidelines. We offered CNY 45 to all participants as compensation.

### 3.2. Measures

#### 3.2.1. Psychological Capital

Psychological capital was assessed using the 24-item Psychological Capital Questionnaire (PCQ) [13]. The PCQ consists of four subscales: self-efficacy, hope, optimism and resilience. Response options ranged from 1 (strongly disagree) to 6 (strongly agree). Sample items included “I feel confident in presenting my work in meetings with management,” “I feel confident in analyzing a long-term problem to find a solution,” “If I should find myself in a jam at work, I could think of many ways to get out of it,” and “When I have a setback at work, I have trouble with recovering from it and moving on.”. Higher values indicated a higher level of psychological capital and its components. The Chinese version of the PCQ has been used in Chinese studies, and it has satisfactory reliability and validity [33,39,40,41]. In the present study, Cronbach’s alpha coefficients for the total scale was 0.79.

#### 3.2.2. Work Engagement

The Job Engagement Scale (JES) was used to measure work engagement [42]. The JES is a self-report questionnaire comprised of 18 items including three subscales: Physiological engagement, cognitive engagement and emotional engagement. The response options were rated on a five-point scale ranging from 1 (strongly disagree) to 5 (strongly agree), with higher scores indicating higher levels of work engagement. The JES was widely used in various occupational groups, and it has satisfactory reliability and validity [15,43,44]. In the current study, the Cronbach’s alpha coefficient for the JES was 0.86.

#### 3.2.3. Safety Citizenship Behavior

Safety citizenship behavior was evaluated using the Chinese version of the Safety Citizenship Behavior Scale (SCBS), which was revised based on the Safety Citizen Role Definition and Behavior Items [3,9]. The SCBS consists of 6 subscales and a total of 27 items: helping, voice, stewardship, whistle blowing, civic virtue, and initiating safety-related change. These items were questions asking the respondents about how much of the described behaviors they believe are part of their job (role-required behavior) or above and beyond their job responsibilities (extra-role behavior). All items were scored on a 4-point Likert scale ranging from 1 (part of the job) to 4 (definitely above and beyond the job), and the total score of all the items was calculated. A higher score indicates a higher-level of activity tendency for the safety citizenship behavior. Sample items were “Volunteering for safety committees”; “Making safety-related recommendations about work activities”; “Protecting fellow crew members from safety hazards”; and “Explaining to other crew members that I will report safety violations”. The Chinese version of the SCBS has been applied among Chinese employees with good reliability and validity [45]. In this study, Cronbach’s alpha coefficients for the SCBS was 0.88.

### 3.3. Statistical Analysis

Correlations analyses and the test of mediation effect were performed using SPSS 21.0 and Mplus 7. The causal steps approach described by Baron and Kenny and the bootstrap method were used in this study to assess the proposed mediation models shown in Figure 1 [46,47]. This process was divided into three steps:

1.Independent variable (psychological capital)–dependent variable (safety citizenship behavior; c-path): proving a significant relationship between the dependent variable (safety citizenship behavior) and the independent variable (psychological capital) is required. Logistic regression analyses should be conducted to test the relationship between the variables in the following model where X represents the independent variable (psychological capital); Y represents the dependent variable (safety citizenship behavior); and M is the mediator (work engagement). The coefficient c is also called the total effect:

**Y = cX + e1.**(1)

2.Independent variable (psychological capital)–mediator (work engagement; a-path): demonstrating a significant relationship between the independent variable (psychological capital) and the hypothesized mediating variable (work engagement). It means that the coefficient a in the regression model must be significant:

**M = aX + e2.**(2)

3.Independent variable (psychological capital)–dependent variable (safety citizenship behavior; c′-path) + mediator (work engagement; b-path): demonstrating a significant relationship between the mediator (work engagement) and the dependent variable (safety citizenship behavior) when both independent variables (psychological capital) and the mediator (work engagement) are predictors of the dependent variable (safety citizenship behavior). Consequently, coefficient b in the regression model must be significant. The coefficient c′ is also called the direct effect, and the product a*b is called the indirect effect. As Preacher and Hayes suggested, the bootstrap method yields the most accurate confidence intervals (CI) for indirect effects [48]. If the 95% CI for the estimates of the mediation effect does not include zero, it suggests that the indirect effect is statistically significant at the 0.05 level. In addition, if coefficient c′ is significant in the present regression model, this means that the mediation effect is a partial mediation effect. Otherwise, the mediation effect is a full mediation effect. The mediated proportions were calculated as the mediation effect divided by the total effect ([a*b]/c):

**Y = c′X + bM + e3.**(3)

## 4. Results

### 4.1. Correlations Analyses

The results of the correlation analyses between the investigated variables were presented in Table 1. As expected, there was a significant positive correlation between the psychological capital and safety citizenship behavior, between the psychological capital and work engagement, as well as between the work engagement and safety citizenship behavior. Thus, according to Baron and Kenny [46], and MacKinnon et al. [49], the results of the correlation analyses among psychological capital, work engagement and safety citizenship behavior meet the requirements to conduct the mediation effect test.

### 4.2. Test the Mediation Effect of Work Engagement

Step 1: Psychological capital–safety citizenship behavior (c-path): as shown in Table 2, psychological capital was significantly associated with safety citizenship behavior (*β* = 0.52, *p* < 0.001), thus supporting Hypothesis 1. The total effect (c) is 0.55. In accordance with the hypothesis, psychological capital was found to be a predictive factor for safety citizenship behavior. This finding is consistent with previous studies which report that psychological capital is an important predictor of job performance and organizational citizenship behavior in general employee groups [15,16,17,18].

Step 2: Psychological capital–work engagement (a-path): as described in Table 3, psychological capital was significantly associated with work engagement (*β* = 0.17, *p* < 0.001). Therefore, Hypothesis 3 was supported. This finding is similar to those of many former studies in which individuals with higher psychological capital have been shown to display better work engagement [30,31,32,33], and further psychological capital plays a very important role in the coal miners’ work engagement.

Step 3: Psychological capital–safety citizenship behavior (c′-path) + work engagement (b-path): as shown in Table 4, significant associations with work engagement were found for safety citizenship behavior (*β* = 0.32, *p* < 0.001), while significant associations with psychological capital were found for work engagement and safety citizenship behavior (*β* = 0.46, *p* < 0.001, support Hypothesis 2). This finding is also in accordance with previous studies that reported that work engagement was significant positively correlated with organizational citizenship behavior [23,24,25,26,27,28,29]. In addition, as shown in Table 5, the results of the bootstrap analysis (a bootstrap sample of 1000 was specified) did not include zero. Thus, Hypothesis 4 was supported because the work engagement mediated the relationship between psychological capital and safety citizenship behavior. Meanwhile, because the coefficient c′ is also significant in the present regression model, the mediation effect is a partial mediation effect. The direct effect (c′) is 0.49. The indirect effect is (a*b) 0.06. The mediated proportion is 12.20%. This result shows that the psychological capital, on the one hand, directly impacts the individual’s safety citizenship behavior, and on the other hand, through the impact on the individual’s work engagement, indirectly affects the individual’s safety citizenship behavior. Therefore, psychological capital plays a decisive role for both work engagement and safety citizenship behavior.

## 5. Discussion

The present study explored the role of work engagement in the relationship between psychological capital and safety citizenship behavior in Chinese coal miners. The relationship between psychological capital and safety citizenship behavior is manifested by the fact that the coal miners with higher psychological capital show a better level of safety citizenship behavior in their organization and work, and the self-efficacy, optimism, hope and resilience of psychological capital all play an important role in the individual’s work.

Self-efficacy refers to the individual’s speculation and judgment about their ability to complete the behavior. People with high self-efficacy tend to choose the tasks that are suitable for their abilities and are challenging, while those with low self-efficacy tend to do the opposite. A person with greater self-efficacy in terms of a particular aspect can expect a greater likelihood of success, be more willing to engage in activities related the aforementioned aspect and enjoy the new behavior for a longer duration; in contrast, they will avoid activities beyond their abilities in which their behavior adherence would also be worse—self-efficacy is therefore one of the decisive factors of human behavior. It is not difficult to understand why the coal miners with higher self-efficacy show a better level of safety citizenship behavior in the organization.

Optimism is one of the most central concepts in positive psychology. According to attribution theory, optimism is defined as an explanatory style obtained through learning [50]. The individual with optimistic attribution in the face of failure often shows positive response attitudes and behavior and will not become frustrated, give up, or admit defeat, and in the face of success will show no pride, continue to work, or transcend themself. On the contrary, the pessimist usually thinks that success is based on luck and that negative experiences and issues are permanent [51]. Therefore, regardless of failure or success, optimism plays a positive role for the individual and optimistic coal miners in the organization show better safety citizenship behavior.

According to Snyder’s hope theory [52], hope is defined as a positive motive state based on the interaction between success, path and willpower, in which willpower is the determination to accomplish the goal, and path refers to the method, strategy, or ability to achieve the goal [52,53]. This means that, in the event of a “road block”, the individual can creatively find an alternative path to accomplish their goal. Therefore, only the staff full of hope can show better safety citizenship behavior in the organization and work.

Resilience refers to the ability of individuals to cope with negative events such as stress, frustration and trauma in life and work [54,55]. Individuals with high resilience can respond effectively and flexibly to the changes in the external environment and can quickly recover from the effects of negative events and maintain a relatively stable and positive life and working state [56]—which is extremely important for coal miners. Because of their complex working environment, high workload and the repetitive tasks entailed, coal miners are prone to fatigue or even job burnout, and once human errors transpire during underground operation, consequences will be disastrous. Therefore, for both coal miners as well as other high-risk occupational groups, it is of great significance for the staff themselves and the whole organization to enhance individuals’ psychological capital so as to effectively promote the individual’s safety citizenship behavior.

We also found that work engagement was significantly associated with coal miners’ safety citizenship behavior. Physiological engagement is not only the basis of safety citizenship behavior, but also a basic requirement of all work behavior of the individual. Without basic physiological engagement, the individual cannot complete the basic content of daily work in the organization or meet basic work requirements, let alone attain the safety objective-oriented organizational citizenship behavior. Of course, in addition to physiological engagement, the staff’s cognitive and emotional engagement in their work is also very important for the individual’s safety citizenship behavior. Emotional engagement in the individual’s work plays a key role in safety assistance behavior, safety advice behavior and safety management behavior. The above three kinds of behavior have a common point, that is, acting as the starting point and the foothold for the others in the organization, and in other words, from a certain point of view, these three kinds of behavior are three different ways of performing social assistance behavior within the organization. The first one is direct assistance behavior, and the latter two are indirect assistance behavior. A large number of previous studies have shown that the individual’s assistance behavior is easily affected by emotional factors in most cases. Likewise, the research also proves the same situation in the organization. In addition, for safety disclosure behavior, safety moral behavior and safety spontaneous behavior, the individual’s cognitive engagement in work is particularly important. In contrast to the three previous kinds of behavior that point towards the behavior of others, these three kinds, whether it is from the starting point or the foothold, point to the individual themself. Therefore, only the individual can fully recognize their position and role in the organization and accurately grasp the goals, values and meanings of their work in order to be more comprehensive and proactive in understanding what responsibilities and tasks to undertake. Therefore, the promotion of work engagement is an important way for the organization to help staff to achieve safety citizenship behavior.

In addition, it is not difficult to determine that work engagement plays an indirect intermediary role between psychological capital and safety citizenship behavior in the study of the relationship between psychological capital, work engagement and safety citizenship behavior in combination with the theoretical basis of previous studies [30,31,32,33]. The results show that psychological capital plays a decisive role for the two other factors. Additionally, work engagement and safety citizenship behavior are important factors that affect the work performance of the organization. Especially for high-risk industries such as coal companies, safety production is the top priority. Therefore, how to effectively enhance the individual’s work engagement and safety citizenship behavior has become a key factor determining whether the enterprise can achieve safety production. Of course, it is a good idea to start from psychological capital. With regard to improving the psychological capital of the staff, attention should be paid to two aspects. On the one hand, enterprises should attach importance to the examination of the psychological capital of candidates in the selection or recruitment process. Enterprises should try to choose the individuals who have better self-efficacy, optimism, hope and psychological toughness, or look for those who have high plasticity in the above aspects. On the other hand, enterprises should strengthen the psychological capital training of existing staff, such as quality development training, psychological training, psychological counseling, group counseling and meditation and other more scientific psychological capital training means to improve individuals’ psychological capital level so as to initially provide the possibility of improving individual work engagement and safety citizenship behavior.

To sum up, the present study provides substantial insight into a complicated interplay among psychological capital, work engagement and safety citizenship behavior in Chinese coal miners. These findings highlight a previously unidentified mechanism explaining the mediator role of work engagement in the relationship between psychological capital and safety citizenship behavior. The case study of Chinese coal miners provides evidence for the external validity of psychological capital as a predictor of safety citizenship behavior. In consideration of the probable mechanisms, it provides some discussion and guidance on how to implement psychological interventions aimed at enhancing coal miners’ safety citizenship behavior. However, an important limitation of the present study must be considered: the present study was a cross-sectional design. Therefore, for further studies, longitudinal studies would provide additional insights into relationships between psychological capital, work engagement and safety citizenship behavior.

## 6. Conclusions

The most salient finding of this study pertains to the integrated model in which work engagement mediates the relationship between psychological capital and safety citizenship behavior. Psychological capital not only has a direct impact on coal miners’ safety citizenship behavior, but also has an indirect impact on coal miners’ safety citizenship behavior via work engagement.

## Figures and Tables

**Figure 1 ijerph-18-09303-f001:**
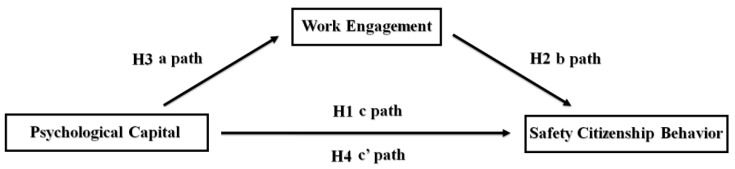
The theoretical model.

**Table 1 ijerph-18-09303-t001:** Correlations of all variables.

Variables	1	1.1	1.2	1.3	1.4	2	2.1	2.2	2.3	3	3.1	3.2	3.3	3.4	3.5	3.6
1 PC	-															
1.1 SE	0.73 **	-														
1.2 Ha	0.82 **	0.53 **	-													
1.3 R	0.83 **	0.54 **	0.61 **	-												
1.4 O	0.77 **	0.38 **	0.51 **	0.45 **	-											
2 WE	0.17 **	0.08	0.11	0.17 **	0.16 **	-										
2.1 PE	0.20 **	0.13 *	0.15 **	0.18 **	0.16 **	0.87 **	-									
2.2 CE	0.10	0.02	0.05	0.12 *	0.11	0.91 **	0.69 **	-								
2.3 EE	0.16 **	0.08	0.08	0.16 **	0.16 **	0.88 **	0.62 **	0.72 **	-							
3 SCB	0.52 **	0.35 **	0.45 **	0.46 **	0.37 **	0.40 **	0.35 **	0.36 **	0.36 **	-						
3.1 Hb	0.48 **	0.34 **	0.46 **	0.43 **	0.31 **	0.34 **	0.31 **	0.32 **	0.27 **	0.84 **	-					
3.2 V	0.49 **	0.32 **	0.42 **	0.45 **	0.35 **	0.31 **	0.30 **	0.27 **	0.25 **	0.81 **	0.64 **	-				
3.3 S	0.45 **	0.30 **	0.34 **	0.42 **	0.33 **	0.31 **	0.27 **	0.27 **	0.29 **	0.81 **	0.63 **	0.67 **	-			
3.4 W	0.45 **	0.31 **	0.41 **	0.35 **	0.34 **	0.30 **	0.24 **	0.26 **	0.29 **	0.85 **	0.63 **	0.59 **	0.64 **	-		
3.5 CV	0.17 **	0.07	0.15 **	0.20 **	0.09	0.34 **	0.28 **	0.33 **	0.29 **	0.64 **	0.40 **	0.39 **	0.42 **	0.45 **	-	
3.6 ISRC	0.38 **	0.28 **	0.31 **	0.31 **	0.28 **	0.32 **	0.29 **	0.27 **	0.30 **	0.79 **	0.57 **	0.53 **	0.56 **	0.61 **	0.48 **	-

Notes: * *p* < 0.05, ** *p* < 0.01; PC = psychological capital, SE = self-efficacy, Ha = hope, R = resilience, O = optimism, WE = work engagement, PE = physiological engagement, CE = cognitive engagement, EE = emotional engagement, SCB = safety citizenship behavior, Hb = helping, V = voice, S = stewardship, W = whistle blowing, CV = civic virtue, ISRC = initiating safety-related change.

**Table 2 ijerph-18-09303-t002:** Results of the regression analysis of the c-path (psychological capital–safety citizenship behavior).

Dependent Variable	Independent Variable	Regression Model	R^2^	B	SE	β	T
Safety citizenship behavior	Psychological capital	Y = cX + e1	0.27	0.55	0.05	0.52	10.68 **

Notes: ** *p* < 0.01, B = regression coefficients, SE = standard error.

**Table 3 ijerph-18-09303-t003:** Results of the regression analysis of the a-path (psychological capital–work engagement).

Dependent Variable	Mediator	Regression Model	R^2^	B	SE	β	T
Work engagement	Psychological capital	M = aX + e2	0.03	0.21	0.07	0.17	3.14 **

Notes: ** *p* < 0.01.

**Table 4 ijerph-18-09303-t004:** Results of the regression analysis of the c′-path and b-path (psychological capital–work engagement + safety citizenship behavior).

Dependent Variable	Independent Variable + Mediator	Regression Model	R^2^	B	SE	β	T
Safety citizenship behavior	Psychological capital	Y = c′X + bM + e3	0.37	0.49	0.05	0.46	10.08 **
	Work engagement			0.29	0.04	0.32	7.09 **

Notes: ** *p* < 0.01.

**Table 5 ijerph-18-09303-t005:** Results of the mediation test (bootstrap analysis for the work engagement of psychological capital on safety citizenship behavior).

Mediated Variable	95% LLCI	Estimated (Indirect Effect)	95% ULCI	Percentage Mediated
Work engagement	0.027	0.06	0.098	12.20

Notes: LLCI = lower limit confidence intervals, ULCI = upper limit confidence intervals.

## Data Availability

The data presented in this study are available upon request from the corresponding author.

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
