# Peer review of "The Role of Work Engagement in the Association between Psychological Capital and Safety Citizenship Behavior in Coal Miners: A Mediation Analysis"

_ijerph, 2021, doi:10.3390/ijerph18179303_

Round 1

Reviewer 1 Report

I congratulate the authors for this very well written article. However, it could benefit from some minor revisions:

Page 3, row 138- the authors should detail in which sense the previous studies of safety citizenship behavior are not adequate

Page 4- Materials and methods- The authors should mention the time period in which the data were collected, the method of recruiting the participants and whether they were offered any reward for participating in the study.

Page 5, row 207- please correct the word “piont”

Author Response

Page 3, row 138- the authors should detail in which sense the previous studies of safety citizenship behavior are not adequate

A: Thank you for your comments. We discussed this problem in details to make this part more clearly.

Page 4- Materials and methods- The authors should mention the time period in which the data were collected, the method of recruiting the participants and whether they were offered any reward for participating in the study.

A: Thank you for your suggestions. We have supplemented those contents in the Materials and Methods.

Page 5, row 207- please correct the word “piont”

A: We have corrected the word. Thanks again for your effort and contribution to the current manuscript.

Reviewer 2 Report

This paper studied the role of work engagement by analyzing the correlation between psychological capital and safety citizenship behavior in mining in China. I agree with the originality and approaching method of the paper, and please modified the following things.

(1) Introduction: The overall dictation is too long and needs to be shortened.

 1) Line 42: Please rewrite the sentence to smooth the connection between the paragraphs. (i.e. Although, attached by ?)

 2) Research purpose: Please create a new paragraph to clearly express the research purpose. In the abstract, the novelty and research purpose of the study are listed.

(2) Theoretical Back Background (section 2)

 1) It is recommended to create a new section and replace 1.1 to 1.3 with section 2.

(3) Materials and Methods

 1) Line 74: Is Age an important factor in this study? If so, please add an analysis of the association between age and the study ?

 2) Line 78: Remove the sentence (Ethics approval ~~~)

 2) It is recommended that the questionnaire used for analysis of PCA, JES, and SCBS be added to supplementary information.

 3) PCA, JES, and SCBS analysis: For the analysis of the study, the methods used in China were used, but is there any difference compared to the international method?

(4) Discussion

  1) Some content in the Discussion recommends moving to results. In discussion, it is necessary to clearly distinguish the difference in results, but they are mixed, which makes the discussion too long.

Author Response

(1) Introduction: The overall dictation is too long and needs to be shortened.

A: Thank you for your comments. We reconstructed the related section to make it more concise.

 1) Line 42: Please rewrite the sentence to smooth the connection between the paragraphs. (i.e. Although, attached by ?)

A: Thank you for your suggestion. We revised this sentence to make it more smoothly.

 2) Research purpose: Please create a new paragraph to clearly express the research purpose. In the abstract, the novelty and research purpose of the study are listed.

A: We have added a paragraph in the end of the literature review section to introduce the research purpose.

(2) Theoretical Back Background (section 2)

 1) It is recommended to create a new section and replace 1.1 to 1.3 with section 2.

A: We have replaced 1.1 to 1.3 with 2 (Literature Review)

(3) Materials and Methods

 1) Line 74: Is Age an important factor in this study? If so, please add an analysis of the association between age and the study ?

A: Thank you for your question. Age is not the important factor in the study. Besides, we didn’t find the theoretical basis about age in the references which we had reviewed.

 2) Line 78: Remove the sentence (Ethics approval ~~~)

A: Thank you for your comment. We have revised the related part.

 2) It is recommended that the questionnaire used for analysis of PCA, JES, and SCBS be added to supplementary information.

A: Thank you for your suggestion. We provide these questionnaires in the supplementary materials.

 3) PCA, JES, and SCBS analysis: For the analysis of the study, the methods used in China were used, but is there any difference compared to the international method?

A: The methods used in this study have been widely used in the social science field. Refences can be found as following:

  1. Chmiel N, Laurent J, Hansez I. Employee perspectives on safety citizenship behaviors and safety violations. Safety Science, 2017, 93, 96-107. https:// doi.org/10.1016/j.ssci.2017.12.010
  2. Jung, Hyo Sun; Yoon, Hye Hyun. The impact of employees’ positive psychological capital on job satisfaction and organizational citizenship behaviors in the hotel. International Journal of Contemporary Hospitality Management, 2015,27, 1135–1156. doi:10.1108/IJCHM-01-2014-0019
  3. Chu, Y.; Lee, K.; Kim, E.I. Why Victimized Employees Become Less Engaged at Work: An Integrated Model for Testing the Mediating Role of Sleep Quality. Int. J. Environ. Res. Public Health 2021, 18, 8468. https:// doi.org/10.3390/ijerph18168468
  4. Zhai, H.; Li, M.; Hao, S.; Chen, M.; Kong, L. How Does Metro Maintenance Staff’s Risk Perception Influence Safety Citizenship Behavior—The Mediating Role of Safety Attitude. Int. J. Environ. Res. Public Health 2021, 18, 5466. https:// doi.org/10.3390/ijerph18105466

(4) Discussion

  1) Some content in the Discussion recommends moving to results. In discussion, it is necessary to clearly distinguish the difference in results, but they are mixed, which makes the discussion too long.

A: According to your suggestion, we have moved some contents from the section of discussion to results. In addition, we have revised the discussion part to make it more clearly. Thanks again for your effort and contribution to the current manuscript.